# A Long-Term Study on Chemical Compounds and Their Location in Sweet Basil Leaves from Organic and Conventional Producers

**DOI:** 10.3390/foods13030383

**Published:** 2024-01-24

**Authors:** Ewelina Hallmann, Anna Rusaczonek, Ewa Muszyńska, Daniel Ziółkowski, Sebastian Kuliński, Jakub Jasek, Alicja Ponder

**Affiliations:** 1Department of Functional and Organic Food, Institute of Human Nutrition Sciences, Warsaw University of Life Sciences, Nowoursynowska 159c, 02-776 Warsaw, Poland; alicja_ponder1@sggw.edu.pl; 2Bioeconomy Research Institute, Agriculture Academy, Vytautas Magnus University, Donelaicio 58, 44248 Kaunas, Lithuania; 3Department of Botany, Institute of Biology, Warsaw University of Life Sciences, Nowoursynowska 159, 02-776 Warsaw, Poland; anna_rusaczonek@sggw.edu.pl (A.R.); ewa_muszynska@sggw.edu.pl (E.M.); 4Military University of Technology, gen. Sylwestra Kaliskiego 2, 00-908 Warsaw, Poland; daniel.z.socrates@outlook.com; 5The University of the West Indies, Cave Hill Rd, Box 1341, Wanstead BB11000, Barbados; sebastian.k.socrates@outlook.com; 6Independent Public Health Care, Henryka Sienkiewicza 7, 09-100 Płońsk, Poland; kuba66714@gmail.com

**Keywords:** antioxidant activity, basil, carotenoids, chlorophylls, conventional, enzymes, organic, polyphenols

## Abstract

Bioactive compound profiles in organic and conventional sweet basil were analyzed by HPLC, and the enzymatic status and antioxidant status of plants cultivated with the two systems were also examined. Fluorescence microscopy was used for the determination of compounds’ locations in the basil leaves. The experiment was conducted from 2019 to 2021. Organic and conventional basil samples were obtained directly from Polish herb producers. The results showed that the chemical profiles of organic and conventional basil leaves are different. Not only the cultivation method but also the experimental year had a significant impact on the antioxidant content in basil leaves. Organic basil contained significantly more dry matter (11.97 g 100 g^−1^ FW) compared to conventional one (10.54 g 100 g^−1^ FW) and a higher tendency for total phenolic compounds (5.24 mg g ^−1^ DW) accumulation. The higher bioactive compound content reflects the antioxidant activity (61.0%, 54.33%, and 46%) in organic basil compared to conventional (46.87%, 38.055, and 39.24%) with respect to the analysis method (ABTS, DPPH, and FRAP). Catalase activity (39 µmol H_2_O_2_ min^−1^ mg^−1^) in organic basil was higher compared to conventional (23.19 µmol H_2_O_2_ min^−1^ mg^−1^) ones. The obtained results are very unique and could be used by herb producers as a key for high-quality basil production. The higher concentration of bioactive compounds in organic basil gives a better nutraceutical status to this popular herb.

## 1. Introduction

Sweet basil (*Ocimum basilicum* L.) is one of the most popular aromatic, medicinal, and spice plants around the world. The plant is also called great basil and is a culinary herb of the *Lamiaceae* family. In Western cuisine, the generic term “basil” refers to the variety also known as sweet basil or Genovese basil [1]. Basil is native to tropical regions from Central Africa to Southeast Asia in temperate climates and is treated as an annual plant; however, basil can be grown as a short-lived perennial or biennial in warmer horticultural zones with tropical or Mediterranean climates [2]. Basil is plant-rich in different bioactive compounds, such as phenolic acids, flavonoids, carotenoids, and chlorophylls. In basil leaves, different phenolic acids were identified: rosmarinic, gallic, caffeic, chlorogenic, hydroxybenzoic, vanillic, ferulic, and trans-cinaminic [3,4]. In the group of flavonoids, rutin, naringin, naringenin, and quercetin were found [5,6]. Green and purple basil leaves contain many pigments. Chlorophylls, protective carotenoids, and anthocyanins are active in the photosynthetic process [7,8]. Basil leaves contain many essential oils. The most important oil compounds are linalool (52.4%), methyl eugenol (18.7%), and borneol (9.3%). The minor compounds are neral (8.1%), 1,8-cineol (5.6%), and myrcene (5.4%) [9,10,11,12]. Such a high concentration of different active substances gives basil phenomenal medicinal, pharmaceutical, and health-promoting properties. The flavonoid composition in *Ocimum basilicum* can boost the phagocytic action of neutrophils and immunostimulant effects [13]. Anti-inflammatory activity and anti-proliferative activity of the hydroethanolic extract of lemon basil (*O. x citriodorum*) were reported in four different human cancer cell lines (HT-144, MCF-7, NCI-H460, and SF-268) [14]. Interestingly, in vitro and in vivo experiments showed that basil is a potent anticancer agent [15]. In another study involving a variety of basil extracts, different compositions with regard to their anticancer activities on several human cancer lines were investigated, so as to verify which combination had the greatest impact [16].

The quality and quantity of chemical compounds in basil depend not only on plant genetics but also on farm management. Organic agriculture is characterized by strict rules. Chemical plant protection and artificial mineral fertilizers are not allowed. Only natural fertilizers, such as plant and animal manure, are widely used [17]. Phenolic compounds are essential elements of plant protection systems. In such situations, plants produce phenolic compounds in response to environmental biotic and abiotic stresses. The major biotic stress of organic plants is pest attack [18,19,20,21]. Therefore, organic plants produce more phenolic compounds [19]. In the present literature, there is still not enough information about organic herbs and spices and their chemical composition. Organic plant management is represented by systems that are stabilized for years. To better understand the chemical composition of organic and conventional plants, long-term experiments should be conducted [22,23,24,25,26]. In the present literature, only a few experiments were conducted with the distribution of phenolic and carotenoid compounds in plant tissues, but not with basil, especially in comparison between organic and conventional plants. Because the organic cultivation system yields different quantities in plant composition, we want to investigate the distribution of compounds in plant tissues [27,28]. The main aim of the present experiment was to show differences in the chemical composition of organic and conventional basil in three years of cultivation. Future results will be very useful information for producers of herbs and spices. They will also indicate to the consumer which basil is worth choosing so that it is a rich source of antioxidant compounds in the daily diet.

## 2. Materials and Methods

### 2.1. Chemicals and Reagents

The following reagents were used in this study: ABTS^•+^ (2′2-azinebis-3-ethylbenzothiazolin-6-sulfonic acid), Sigma-Aldrich Company (Poznan, Poland); acetonitrile (HPLC grade), Sigma-Aldrich Company (Poznan, Poland); acetone (HPLC grade), Sigma-Aldrich Company (Poznan, Poland); aluminum chloride (pure for analysis grade), Merck (Poland, Poznan); ammonium molybdate (pure for analysis grade), Sigma-Aldrich Company (Poznan, Poland); carotenoid standards (beta-carotene, lutein, zeaxanthin) and chlorophyll standards (chlorophyll a, chlorophyll b), Sigma-Aldrich Company (Poznan, Poland); deionized water and ethyl acetate (HPLC grade), Merck (Poznan, Poland); Folin–Ciocalteu reagent (pure for analysis grade), Sigma-Aldrich Company (Poznan, Poland); methanol (HPLC grade), Sigma-Aldrich Company (Poznan, Poland); hydrochloric acid (35%), Sigma-Aldrich Company (Poznan, Poland); PBS (phosphate-buffered saline), Sigma-Aldrich Company (Poznan, Poland); potassium persulfate (pure for analysis grade), Sigma-Aldrich Company (Poznan, Poland); sodium hydroxide (pure for analysis grade), Merck (Poznan, Poland); sodium carbonate (pure for analysis grade), Merck (Poznan, Poland); polyphenols standards (benzoic acid, caffeic acid, ferulic acid, gallic acid, kaempferol, kaempferol-3-O-glucodside, luteolin, myricetin, quercetin, quercetin-3-O-glucodside), Sigma-Aldrich Company (Poznan, Poland); hydrogen peroxide solution (>30%, for trace analysis), Merck (Poznan, Poland); trichloroacetic acid (BioXtra), Sigma-Aldrich Company (Poznan, Poland); potassium phosphate dibasic (ACS reagent), Sigma-Aldrich Company (Poznan, Poland); potassium dihydrogen phosphate (for analysis), Sigma-Aldrich Company (Poznan, Poland); potassium iodide (BioXtra), Sigma-Aldrich Company (Poznan, Poland); TPTZ (2,4,6-tripyridyl-s-triazine), Merck (Poznan, Poland); potassium peroxodisulfate (ACS reagent), Sigma-Aldrich Company (Poznan, Poland); DPPH^•^ (2,2-diphenyl-1-picrylhydrazyl), Merck (Poznan, Poland); Bradford reagent, Sigma-Aldrich Company (Poznan, Poland); hydrochloric acid, Sigma-Aldrich Company (Poznan, Poland); iron(III) chloride, Sigma-Aldrich Company (Poznan, Poland); tricine (titration), Merck (Poznan, Poland); ethylene glycol-bis(2-aminoethylether)-N,N,N′,N′-tetraacetic acid (molecular biology grade), Sigma-Aldrich Company (Poznan, Poland); magnesium sulfate solution (molecular biology grade), Sigma-Aldrich Company (Poznan, Poland); DL-dithiothreitol (molecular biology grade), Sigma-Aldrich Company (Poznan, Poland); prediluted protein assay standards: bovine serum albumin set, Merck (Poznan, Poland); (-)-riboflavin (reagent grade), Sigma-Aldrich Company (Poznan, Poland); nitrotetrazolium blue chloride (HPLC grade), Sigma-Aldrich Company (Poznan, Poland); L-methionine (reagent grade), Sigma-Aldrich Company (Poznan, Poland); ethylenediaminetetraacetic acid disodium salt dihydrate (molecular biology grade), Merck (Poznan, Poland).

### 2.2. Equipment

The following equipments were used in this study: Drier Farma Play FP-25 W (Bielsko-Biala, Poland), TissueLyser LT (Qiagen, Eindhoven, The Netherlands), Multiscan-GO microplate reader from Thermo Scientific (Rockford, IL, USA), LED lamp (SL3500-W-D; PSI), ultrasonic bath (Polsonic, Szczecin, Poland), BioSENS UV-6000 spectrophotometer (Warsaw, Poland), centrifuge Hermle Z 300K (Wrocław, Poland); HPLC setup was used, consisting of two LC-20AD pumps, a CMB-20A system controller, a SIL-20AC autosampler, an ultraviolet-visible SPD-20AV detector, a CTD-20AC oven (Shimpol, Warsaw, Poland), vaccum-pump AgaLabor (Warsaw, Poland), and fluorescence microscope equipped with a UMNU narrow-band filter cube (Olympus-Provis, Tokyo, Japan).

### 2.3. Plant Material

The experiment was carried out in three continuous years: 2019–2021. Basil samples cv. “Genovense” were obtained directly from organic and conventional herb and spice producers in Poland (Figure 1). Both organic and conventional sweet basil seeds were purchased directly from the largest seed producer Plantico, Poland. Seeds were delivered to herb producers. Herbs were cultivated in greenhouses in organic and conventional conditions according to law rules (regulation no. 2018/834) [29]. Basil was cultivated in organic and conventional farms (producers). The seeds were sown every year in April (15 April 2019, 17 April 2020, and 14 April 2021) in organic coconut-fiber multiplied pallets (in organic farms) and plastic multiplied pallets (in conventional farms). Organic production was carried out with compost use, as a basic growing medium. Conventional production was carried out with peat as a basic growing medium. Seeds were watered to keep the required growing medium moisture. Twenty days after sowing, the seedlings were transplanted into 1.0 L pots, containing an appropriate growing substrate with fertilizers. In organic farms, producers use Humvit bio, Humiplant, Algaplant, and Humvit bio universal. In conventional farms, producers use Ziołovit universal, Agrolinija-S, and Basofoliar 2.0. In organic farms, no chemical plant protection was conducted. In conventional farms, yellow sticky traps (according to producers’ declaration) were used against dark-winged fungus gnats. The temperature inside the greenhouses was between 20 and 25 °C, and the total humidity was 65–70%. A detailed summary of average values for total radiation and air temperature is presented in Appendix A.

When basil plants grow up to 15 cm, after 6 weeks, plants were used for experimental purposes. In every experimental year, three producers (the same in the whole experiment) were chosen. Six individual pots of basil were purchased from each producer in both cultivation systems (organic and conventional); ORG, *n* = 18 and CONV, *n* = 18 per experimental year. In the whole 3-year experiment, 54 plants (repetitions) were evaluated. Fresh basil was gently transported to the laboratory. One plant was treated as a repetition. For experimental purposes, only basil leaves were used. After leaf separation from the plants, the material was divided into two parts: first only for dry matter measurement and second for freeze-drying. For analytical purposes, plant material was put into liquid nitrogen (−196 °C) and then freeze-dried with a LabconCo freeze-dryer (−60 °C, 0.100 mBar, 96 h). After that, the plant material was powdered with Mill-A11 basic IKA equipment. All results are presented in DW (dry weight) units.

### 2.4. Chemical Analysis

#### 2.4.1. Dry Matter Content

Dry matter was measured by the gravimetric method described in the Polish standard protocol [30]. Briefly, the dry matter content was calculated based on the mass differences before and after drying at 105 °C.

#### 2.4.2. Hydrogen Peroxide Level Determination

To assess hydrogen peroxide (H_2_O_2_) levels, we followed the method originally described with some slight modifications [31]. A powered freeze-dried tissue sample was subjected to homogenization with cold 0.1% trichloroacetic acid. Subsequently, the homogenate was centrifuged, and the supernatant was further mixed with 10 mM potassium phosphate buffer (pH 7.0) and 1 M KI. The absorbance of the resultant mixture was measured at a wavelength of 390 nm. To determine the concentration of H_2_O_2_, we employed a suitable standard curve. The data obtained were expressed as micromoles of H_2_O_2_ per 100 mg of DW.

#### 2.4.3. Protein Extraction and Enzyme Activity Measurements

A powered freeze-dried tissue sample was effectively homogenized, along with extraction buffer containing 100 mM tricine, 3 mM MgSO_4_, 3 mM EGTA, 1 mM dithiothreitol, and 1 M Tris/HCl at a pH of 7.5. Following homogenization, the samples were incubated on ice and then subjected to centrifugation. Subsequently, the protein concentration was ascertained using a Bradford assay kit provided by Thermo Scientific, employing bovine serum albumin (BSA) as the standard. The spectrophotometric assay of superoxide dismutase (SOD) activity was conducted in accordance with the original method, incorporating modifications as previously documented [32,33]. The enzyme assay mixture comprised 0.1 M phosphate buffer at a pH of 7.5, 2.4 μM riboflavin, 840 μM NBT, 150 mM methionine, and 12 mM Na_2_EDTA. The enzyme extract was combined with the enzyme assay mixture in a manner ensuring inhibition of NBT oxidation within the range of 20% to 80%. Absorbance measurements were taken at 560 nm for 15 min following sample exposure to either 500 μmol m^−2^s^−1^ illumination or incubation in darkness as a blank sample. The results were expressed in units, representing the quantity of enzyme required to inhibit NBT photoreduction to blue formazan by 50% per milligram of protein. Catalase (CAT) activity was determined spectrophotometrically, following the methodology outlined by the original method, with some adjustments [34]. Perhydrol was diluted with 50 mM phosphate buffer (pH 7.0) to achieve an absorbance of 0.5 (±0.02) at 240 nm, representing an initial H_2_O_2_ concentration of approximately 13 mM. The enzyme extract was combined with reaction buffer in a manner that resulted in a decrease in absorbance falling within the range of 20% to 80%. CAT activity was quantified based on the rate of H_2_O_2_ degradation observed over a 2 min period, utilizing the molecular extinction coefficient of H_2_O_2_ at 240 nm (ε = 43.6 mol^−1^ cm^−1^). The results were expressed in units of micromoles of H_2_O_2_ per minute per milligram of protein.

#### 2.4.4. Total Polyphenol Content

The total polyphenol content was measured by the Folin–Ciocâlteu method [35]. One hundred milligrams of powdered plant material was weighed into a 250 mL beaker, and 50 mL of 80% methanol was added. The samples were sonicated (20 min, 6000 Hz, temp. 30 °C). The samples were then vacuum-filtered. The obtained supernatant was used for assays. A specified amount of the solution of the tested extracts (1.0 mL) was placed into 50 mL volumetric flasks, then 2.5 mL of Folin–Ciocâlteu reagent and 5.0 mL of 20% sodium carbonate (Na_2_CO_3_) were added, and distilled water was added to the mark. Samples were incubated for 45 min at room temperature in the dark. After this time, absorbance was measured at a wavelength of λ = 750 nm. The total polyphenol content was calculated according to a mathematical formula with a dilution coefficient.
y = (2.125 × (absorbance) + 0.1317) × 100

The results are presented as gallic acid equivalents: GAE mg g^−^^1^ DW.

#### 2.4.5. Total Phenolic Acid Content

Total phenolic acid was measured by the Arnov method [36]. One hundred milligrams of powdered plant material was weighed into plastic testing tubes; next, 5 mL of 80% ethanol was added. Samples were extracted by sonication (10 min, 6000 Hz, temp. 30 °C) and then centrifuged (6000 rpm, 10 min, 0 °C). One milliliter of supernatant was injected into a glass testing tube and 1 mL of 0.5 M chloric acid (35%) HCl, 1 mL of Arnov reagent, 1 mL of 1 M sodium hydroxide, and 6 mL of pure water. The sample was mixed up and down and used for UV-Vis absorption measurements at a wavelength of 490 nm. A standard curve was prepared with gallic acid, and the results were calculated according to a mathematical formula and presented as GAE mg g^−^^1^ DW.
y = (1353.2 × (absorbance) + 3.9761) × 10

#### 2.4.6. Total Flavonoid Content

Total flavonoids were determined by the colorimetric method [37]. One gram of fine plant material was dissolved in methanol (100%). The samples were extracted in an ultrasonic bath and then filtered under vacuum. The obtained extract (5 mL) was mixed with sodium acetate C_2_H_3_NaO_2_ (5.0 mL, 100 g L^−^^1^) and aluminum chloride AlCl_3_ (3.0 mL, 25 g L^−^^1^) and brought to 25 mL with methanol in a calibrated flask. Each solution was compared with the same mixture but without the reagent (AlCl_3_). The absorbance was measured at 425 nm. The total flavonoid content was calculated using the equation obtained from the calibration curve of rutin (quercetin-3-O-rutinoside, Q-3-R), according to the mathematical formula and the results are presented as mg g^−^^1^ DW.
y = (6.452 × (absorbance) + 0.451) × 25

#### 2.4.7. Individual Phenolic Identification and Quantification

Individual phenolics were extracted and measured by HPLC [38]. One hundred milligrams of powdered plant material was extracted with 80% methanol by sonication (10 min., 6000 Hz, temp. 30 °C). After extraction, the samples were centrifuged (10 min, 6000 rpm, temp. 0 °C). Fifty microliters of supernatant was injected into a Fusion RP-80 A column (250 × 4.6 mm, Phenomenex, Warsaw, Poland). A flow rate of 1 mL min^−1^ of gradient phase prepared from acetonitrile and water with phosphoric acid (pH 3.0) was used. Pump pressure was in the range of 13.00–14.50 mPa. Time phases are as follows: 1.00–22.99 min, phase A 95% and phase B 5%; 23.00–27.99 min, phase A 50% and phase B 50%; 28.00–28.99 min, phase A 80% and phase B 20%; and 29.00–38.00 min, phase A 95% and phase B 5%. The analysis time was 42 min, the detection wavelength for flavonoids was 360 nm, and the detection wavelength for phenolic acids was 250 nm. Polyphenols were identified based on retention time and external standards (gallic acid, p-hydrobenzoic acid, caffeic acid, p-coumaric acid, benzoic acid, kaempferol-3-O-glucoside, myricetin, luteolin, quercetin, quercetin-3-O-glucoside, and kaempferol). Chromatograms from phenolics analysis are presented in Appendix A. Standard curves with their equations and R^2^ are presented in Appendix A.

#### 2.4.8. Total Chlorophyll and Carotenoid Contents

Total chlorophyll and carotenoid contents were measured by the colorimetric method [39]. A 100 mg sample of examined plant material was weighed into a glass beaker. Next, 50 mL of cold acetone (−20 °C) was added to the weighed sample and thoroughly macerate the sample (mix the extract with a glass rod). The contents of the beaker were transferred to a Schott funnel and paper filter under vacuum until the chlorophyll was completely washed out of the sample (poured cold acetone until the falling drops are colorless). The contents of the receiving flask were quantitatively transferred to a 50 mL volumetric flask and, if necessary, made up to the mark with acetone. The sample was mixed thoroughly, and the absorbance of the test sample was measured at the following wavelengths: 441 nm, 646 nm, and 663 nm (the spectrophotometer was previously calibrated against cold 80% acetone). The extinction coefficients for chlorophyll a and chlorophyll b were calculated, and then, the contents of chlorophyll (a and b) and the sum of carotenoids were calculated using the following formulas:coefficient (chlorophyll a) we calculate from equilibration: (12.21 × A663) − (2.81 × A646)
coefficient (chlorophyll b) we calculate from equilibration: (20.13 × A646) − (5.03 × A663)
Chlorophyll a (µg g^−1^ DW) = (coef. chl. a × 50 × 1 g)/(1000 × 0.1)
Chlorophyll b (µg g^−1^ DW) = (coef. chl. b × 50 × 1 g)/(1000 × 0.1)
Total carotenoids (µg g^−1^ DW)= (1000 × (A441 − ((3.27 × coef. chl. a) − (104 × coef. chl. b))))/229

#### 2.4.9. Individual Carotenoids and Chlorophylls Identification and Quantification

Chlorophylls and carotenoids were measured by the HPLC method [38]. A small amount (50 mg) of powdered plant tissue was extracted with cold acetone, and then, magnesium carbonate was added. The samples were incubated in a cold ultrasonic bath (0 °C, 15 min). After extraction, the samples were centrifuged (5500 rpm, 2 °C, 10 min). And 1 mL of centrifuged extract was used for the next step of analysis. Fifty microliters of supernatant was injected into a Max RP-80 A column (250 × 4.6 mm, Phenomenex, Warsaw, Poland). For analytical purposes, two mobile phases were used. The first mobile phase (A) contained 90% acetonitrile and 10% methanol. Second phase (B) contained 68% methanol and 32% ethyl acetate with flow of 1 mL min^−1^ and time program of 1.00–14.99 min, Phase A 100%; 15.00–22.99 min, Phase A 40% and Phase B 60%; and 24.00–28.00 min, Phase A 100%. The wavelengths used for detection were 450 nm. The carotenoids and chlorophylls were identified based on Fluka and Sigma Aldrich external standards (lutein, zeaxanthin, beta-carotene, chlorophyll a, chlorophyll b). Chromatograms from carotenoids and chlorophylls identification are presented in Appendix A.

#### 2.4.10. Antioxidant Activity: ABTS, DPPH, and FRAP

The antioxidant potential of methanol extracts was analyzed by the scavenging ability of ABTS and DPPH free radicals and the ability to reduce Fe^3+^TPTZ to Fe^2+^TPTZ (FRAP assay). A powered freeze-dried tissue sample was homogenized with methanol and centrifuged (15 min, 13,000 rpm, RT). The supernatant was used to determine the antioxidant potential of the samples. The activity against ABTS^•+^ was investigated as previously described, following the original method [40]. To obtain the green-blue ABTS^•+^ solution, an aqueous mixture of 7 mM ABTS (2,2′-azinobis(3-ethylbenzothiazoline-6-sulfonic acid)) and 2.45 mM potassium persulfate was left to stand in the dark overnight at room temperature. Later, the ABTS^•+^ solution was diluted with ethanol until the absorbance was 0.7 ± 0.02 at 734 nm. The supernatant of the prepared samples was further mixed with ABTS^•+^ solution, and after 6 min of reaction, the absorbance was measured at a wavelength of 734 nm. The percentage inhibition of ABTS^•+^ by the samples was calculated. The activity against DPPH radicals was examined as per the previously described method, following the original procedure [41]. The supernatant of the prepared samples was mixed with an ethanol solution of DPPH^•^ (2,2-diphenyl-1-picrylhydrazyl), and after 20 min of reaction, the absorbance was measured at a wavelength of 517 nm. The percentage inhibition of DPPH^•^ by the samples was calculated. One equation was used for ABTS and DPPH methods to determine the antioxidant activity, as follows: (%) = [(A0 − A1)/A0] × 100, where A0 and A1 are the absorbance intensities of the control and the sample, respectively. Activity expressed as the ability to reduce Fe^3+^ TPTZ (2,4,6-tripyridyl-s-triazine) to Fe^2+^TPTZ was performed according to the FRAP assay [42]. Briefly, the working FRAP solution was freshly prepared by adding 10 mM TPTZ in 40 mM HCL, 20 mM FeCl_3_, and 300 mM acetate buffer (pH 3.6) at a ratio of 1:1:10. The supernatant of the prepared samples was mixed with FRAP solution, and the absorbance was measured at a wavelength of 593 nm using a microplate reader. The percentage reduction of Fe^3+^ TPTZ to blue-colored Fe^2+^TPTZ by the samples was calculated as follows: (%) = (A1/A0) × 100, where A0 and A1 are the absorbance intensities of the reference standard (ascorbic acid) and the sample, respectively.

#### 2.4.11. Microscopic Analyses

The localization of secondary metabolite autofluorescence was performed according to the original method [43]. Handmade cross-sections of fresh samples were prepared from the leaves located in the second node on the stem, mounted on glass slides in a drop of distilled water, and observed under a bright field or UV excitation. Ten different leaf blades and their petioles were applied for both conventional and organic cultivation.

### 2.5. Statistical Analysis

The results obtained from chemical measurements were statistically evaluated using Statgraphics Centurion 15.2.11.0 software (StatPoint Technologies, Inc., Warrenton, VA, USA). Two-way ANOVA using Tukey’s test (α = 0.05) was used. The factors of the experiment were the basil origin (ORG vs. CONV) and the cultivation year (2019, 2020, and 2021). In the tables, average values with standard deviation for individual factors and interactions between factors are shown. Different letters indicate statistically significant differences between experimental groups. For overall correlation evaluation, a principle component analysis (PCA) was used. The PCA figures were made using XLStat (Microsoft Excel version 16.18).

## 3. Results and Discussion

### 3.1. Dry Matter in Basil

The obtained results showed that organic basil contained significantly more (*p* = 0.0001) dry matter than conventional basil. We also observed the effect of the cultivation year. In the first experimental year, basil contained significantly more (*p* < 0.0001) dry matter than in the next two years. The concentration of dry matter in plants in different cultivation years could be an effect of weather conditions, especially temperature and light. According to weather data (Appendix A), during the first experimental year, the average day/night temperature during the time of intense basil growth was the highest. Similar results were presented in other experiments with sour cherries [26,43,44]. In the case of sour cherries, in the two first experimental years (2015–2016), fruits contain more dry matter compared to the third and fourth experimental years. It was possible to find a link between temperature and sunlight during the fruiting period of trees and dry matter concentration in fruits [43]. The apricots cultivated in organic and conventional orchards in a two-year cycle showed differentiation in the case of dry matter production in fruits. In the second year, fruits contain significantly more dry matter compared to the first one. In 2013, the experimental orchards had much more sun and higher temperatures compared to 2012 (the first experimental year) [44]. The interaction between cultivation and experimental years shows that every year, organic basil was characterized by a significant (*p* = 0.0045) concentration of dry matter (Table 1). Organic practices are affected by higher dry matter concentrations in herbs, such as marjoram and oregano, sage, and lemon balm but not basil [45]. Basil cultivated with organic fertilizers produced more shoot and leaf dry matter in the second experimental year than in the first year of observation [46,47]. In the present experiment, significantly more dry matter was observed in the organic basil in the first and third years of the experiment. Plants cultivated with organic soil growing media showed a higher rate of assimilation and photosynthesis. This can result in higher dry matter concentrations in plant tissues [48]. On the other hand, both organic and conventional plants absorb inorganic ions along with water from the soil. In the case of organic plants, the presence of soil organisms makes ion uptake easier. Therefore, conventional plants pump more water along with ions into their tissues. It changes the balance between dry matter and water content in plant tissues. Many studies have shown that organic leafy vegetables and herbs are characterized by a higher dry matter content than conventional vegetables [49,50].

### 3.2. Enzymatic Status of Plants

The high concentration of hydrogen peroxide (H_2_O_2_) in plants is an effect of oxidative stress. On the other hand, superoxide dismutase (SOD) is one of the key enzymes in the detoxification pathway of H_2_O_2_. In our experiment, we observed that conventional basil was characterized by a significantly (*p* < 0.0001) higher activity of SOD, as well as a significantly higher concentration of H_2_O_2_ (*p* < 0.0001) compared to organic basil. We concluded that the higher concentration of phenolics in organic basil led to a low-stress situation, as we noted a connection with biotic stresses in the organic environment (Table 1). Conventional plants are protected by artificial pesticides at the time of cultivation. They protect against biotic stresses, but chemical agents create abiotic stresses focused on oxidative stress and H_2_O_2_ levels in plant tissues. On the other hand, organic basil was characterized by significantly higher catalase (CAT) activity (*p* < 0.0001) than conventional basil. Perhaps, this enzyme together with polyphenolic compounds is the basis of plant defense against environmental stresses. We also found a strong and significant correlation between years and the cultivation system (*p* = 0.0035). In all experimental years, organic basil was characterized by a significant and higher CAT activity (Table 2). The increase in CAT activity in organic basil could be an effect of salinity stress. Pepper mint reacts perfectly by increasing CAT synthesis over time under salinity stress conditions [51]. Organic cultivation methods consist of feeding plants only with organic fertilizers. Soil life in the presence of water decomposes organic matter into inorganic compounds. It may happen that in the root zone of organic plants, the level of salinity will suddenly increase. In such situations, organic basil plants may respond by increasing the synthesis of the CAT enzyme (Table 2).

### 3.3. Phenolic Compounds in Basil

In the present experiment, we did not observe differences in total polyphenols, phenolic acids, or total flavonoid content between organic and conventional basil. Only the effect of years was noticed. In 2021, plants produced significantly more phenolics (*p* < 0.0001; *p* = 0.0005; *p* < 0.0001) than in the first and second experimental years. The higher concentration of polyphenolic compounds in one of the three years of the experiment (2021) may be the result of higher intensity of sunlight (J m^−2^) during basil growth (Appendix A). Basil cultivated in an open-field cultivation system produced significantly more polyphenols compared to the greenhouse system. This is because more light was available for plants in an open-field system [52]. No significant interaction between the cultivation system and the experimental year was observed in the case of all phenolics (Table 1). The obtained results are contrary to those presented for oregano [19]. On the other hand, thymus plants sprayed with effective microorganisms (EMs) and bioalgeen S90 stimulators showed higher concentrations of total polyphenols and phenolic acids than untreated plants [53]. Organic air-dry basil was characterized by a higher content of all total phenolics (phenolic acids and flavonoids) compared to conventional basil in two experimental years [38]. In our experiment, we observed only some tendency for a higher phenolic concentration in organic plants, but no statistical evidence was obtained (Table 1). To the best of our knowledge, organic plants produce and use phenolics as defense systems against pests and fungal attacks [54]. Evidently, other factors, such as cultivars, weather conditions, or agricultural practices, are also important.

Individual profiling of phenolic acids showed that conventional basil contained significantly more p-coumaric acid (*p* < 0.0001) than organic basil. We also observed a strong correlation between experimental combinations. In three years, conventional basil always contained significantly more p-coumaric acid (*p* < 0.0001). P-coumaric acid is well known for its antioxidant properties. It scavenges reactive oxygen species (ROS) and regulates endogenous antioxidant enzymes, thereby preventing oxidative damage to biomolecules. From a nutritional point of view, p-coumaric acid strengthens the body’s antioxidant defense [55]. P-coumaric acid has strong antibacterial properties against Gram(+) and Gram(-) pathogenic bacteria. Some experiments have shown that p-coumaric acid kills pathogenic strains of bacteria, such as *Escherichia coli*, *Shigella dysenteriae*, and *Salmonella typhimurium* [56]. Therefore, using basil in everyday meal preparation increases the food security of consumers. Organic basil was characterized by a higher p-coumaric acid concentration than hydroponic basil [57]. We obtained opposite results, but not with hydroponic cultivation, but with a typical conventional cultivation system. Another important phenolic acid that is characteristic of basil is rosmarinic acid. In examining basil leaves, we do not observe statistically significant differences between organic and conventional basil. Only the cultivation year had a significant impact on rosmarinic acid concentration. In 2019, basil contained the lowest level of rosmarinic acid (<0.0001), compared to the 2020 and 2021 experimental years (Table 3). A similar situation was observed with the individual compounds mirycetin (*p* < 0.0001) and luteolin (Table 2).

### 3.4. Carotenoid Compounds in Plants

Carotenoids and chlorophylls are compounds essential for the photosynthesis process. Green colorants in plant leaves are directly connected with the nitrogen concentration in plants. In conventional cultivation systems, mineral, easily soluble nitrogen fertilizers are allowed. It affects the higher concentration of nitrogen in plant tissues and stimulates chlorophyll synthesis in leaves [58]. We did not observe a significant effect of the cultivation method or experimental year on the chlorophyll content in basil leaves. Only the interaction between cultivation and year was important (*p* < 0.0001). In every experimental year, conventional basil samples were characterized by a higher chlorophyll concentration in leaves (Table 1). Similar findings were observed with organic and conventional basil and thymus, but opposite results were presented with oregano [19,38,59]. In the individual chlorophyll fraction, only chlorophyll b was present in higher concentrations in the organic basil. Profiling of individual chlorophylls (a and b) showed that in the first two years, organic basil contained a significant (*p* < 0.0001) higher fraction compared to conventional basil (Table 3).

Carotenoids are compounds connected with chlorophyll synthesis. Leafy plants, such as basil, produce photosynthetic system protection against photodestruction. The higher chlorophyll concentration in basil leaves is directly connected with high carotenoid abundance. In our experiment, we found an effect of cultivation (*p* = 0.002) but not year on the total carotenoid concentration in basil. On the other hand, we observed a strong and statistically significant (*p* < 0.0001) interaction between system and year. In every experimental year, conventional basil contained significantly more total carotenoids (Table 1). After carotenoid fraction measurement, we observed that only the lutein concentration was higher and statistically significant in the organic basil (Table 2). Basil is a good source of lutein. In the leaves, 9.23 mg g^−1^ DW of lutein was found [60]. In our experiment, the amount was lower but statistically significant. The organic basil contained significantly (*p* = 0.022) more lutein than the conventional basil, and in 2020, the basil produced significantly more (*p* = 0.0032) lutein than the rest of the experimental years (Table 3). Various observational and interventional studies suggest that lutein has a possible role in reducing the risk of AMD syndrome (age-related macular degeneration). Consumption of food rich in lutein, such as basil, increases protection against this disease [61].

### 3.5. Antioxidant Status in Plants

All compounds belonging to phenolic groups are strong antioxidants. They are stronger when they play an active role as free radical scavengers. As we noted, organic basil had a tendency to have higher polyphenol concentrations than conventional basil (Table 1). The presence of phenolics reflects the antioxidant power of basil. In the present experiment, organic basil was characterized by a significant (*p* < 0.0001) and higher antioxidant activity compared to conventional basil in the case of three different measuring methods (ABTS, DPPH, and FRAP) (Figure 2 and Figure 3). It is worth noting that in the third experimental year (2021), basil was characterized by the highest antioxidant power measured by three different methods. This is in accordance with the concentration of phenolics in plant tissues in that year (Table 1 and Table 2). In experiments conducted over two years with basil, authors showed that antioxidant activity was different between experimental years. In the first year, antioxidant activity and concentration of total polyphenols were much lower compared to the second experimental year [62]. Organic oregano, thyme, and rosemary contained more total polyphenols than conventional ones [63]. At the same time, all tested organic herbs were characterized by significantly higher ABTS, DPPH, and FRAP activity compared to conventional herbs. Similar findings were presented with spinach plants. Organic spinach contains more phenolics compared to conventional ones. Therefore, organic samples were characterized by a higher antioxidant power than conventional spinach [64]. In the present experiment, we found a strong interaction between cultivation methods and years. Always, in three years of basic observation, organic plants were characterized by a higher antioxidant activity, but only the antioxidant activity measured by the ABTS method was statistically significant (*p* = 0.0300).

### 3.6. Principal Component Analysis

The PCA results showed that the overall degree of variability explained by F1 and F2 was 85.32% as shown in Figure 4. This was confirmed by a strong link between the chemical composition of organic and conventional sweet basil and experimental years. Two experimental years, 2019 and 2020, were similar to each other. The experimental objects cultivated in those two years were located in separate areas, but very close to each other. The observations carried out indicate that in 2019–2020, the quality of basil significantly depended on the content of chlorophylls, zeaxanthin and phenolic acids, such as benzoic, ferulic, and gallic (Figure 4). The year 2021 was completely different from the two previous years of research. The experimental objects observed in 2021 were located in a completely different, separated part of the chart. This arrangement suggests a complete chemical dissimilarity between the examined basil plants. In 2021, the quality of basil depended mostly on the highest concentration of flavonoid compounds. The organic and conventional experimental objects were located in different and separate chart areas. Organic basil was characterized by a significant concentration of dry matter, kaempferol, and rosmarinic acid, as well as catalase activity. Conventional basil was characterized by a high concentration of total carotenoids, as well as individual compounds, such as beta-carotene, p-coumaric acid, mirycetin, and lutein (Figure 4).

### 3.7. Fluorescent Identification of Bioactive Compounds in Plants

The presented results showed that both organic and conventional basil produce bioactive compounds from phenolic, carotenoid, and chlorophyll groups. Quantitative and qualitative comparison of the bioactive compounds in basil leaves shows that the cultivation system has a significant effect on the chemical basil composition. In our research, we performed yet another unique test of fluorescent identification of the place of accumulation of bioactive compounds in basil leaves from organic and conventional cultivation. The fluorescence microscopy images showed that there are some visible differences in the localization of biologically active compounds in the leaves of organic and conventional basil (Figure 5). The young leaves from the plants’ tops are similar in both systems: organic and conventional. Some palisade parenchyma cells show orange-yellow autofluorescence; however, it is a phenomenon visible in weak areas because of strong chlorophyll fluorescence. Therefore, tannins, aromatic quinones, and/or alkaloids occur (A1 and A2) in the young cells from that layer. It is worth noting that young leaves from organic basil begin to accumulate compounds in the upper epidermis, whereas conventionally, they do not. Organic basil epidermal cells give a turquoise color. This may be an effect of autofluorescence imposition of compounds with blue and green emission (blue emission may represent coumarins, C6–C3 phenolic compounds, and stilbenes, while green color gives phenolics such as chlorogenic acid and gallic acid). It is also probable that the turquoise color belongs to flavonols quercetin and kaempferol (B1 and B2). Bioactive compounds, including flavonoids and phenolic acids, generally accumulate in the vacuoles of epidermal cells and act to decrease the transmittance of UV wavelengths to the mesophyll, protecting the deeper photosynthetic layers of the leaf (C1 and C2) [65]. Similar findings were presented in another experiment [66]. This cell autofluorescence was probably due to the bathochromic shift of the flavonoids and/or other phenolics contained within the epidermal cells. Kaempferol glycosides are the main compounds in the leaves of *Quercus ilex*, whereas luteolin, apigenin, and quercetin, both glucosides and aglycones, are components of the hairs of olive leaves. In our experiment, the main flavonoids in organic basil leaves were quercetin-3-O-glucosides and kaempferol. In the case of conventional basil, it was luteolin and quercetin (Table 3).

In conventional basil, palisade parenchyma cells of older leaves from the second node (fully developed) and upper epidermis puzzle-shaped cells showed blue autofluorescence of cell sap (B1 and C1). Some cells are more intensely blue, which may suggest a higher concentration of some bioactive compounds. Additionally, in some cells of spongy parenchyma, chloroplasts are visible on an intensely red background. That color is an effect of the autofluorescence of anthocyanins, some flavonoids, and phenolic acids. This is in accordance with qualitative analysis and higher luteolin and p-coumaric acid concentrations in conventional basil. It is worth noting that in the organic basil epidermis, the cells are turquoise, while palisade parenchyma cells are in pink-purple color (B2 and C2). These are the effects of the imposition of red and blue or yellow and red autofluorescence of compounds. Additionally, there are strongly fluorescent cells of spongy parenchyma in a similar color. Leaf petioles in both organic and conventional basil have cuticles covered with cutin with blue autofluorescence (D1 and D2). We can observe below-ground parenchyma cells with dark red anthocyanins. These cells are characteristic of cold/phosphorus stress reactions. Under normal conditions, they are in sleep mode, but in times of stress, they start to be visible in purple. In organic basil petioles, we identified many more cells with compounds showing green autofluorescence (D2). It is the effect of a higher concentration of some phenolics, such as kaempferol (Table 3).

## 4. Conclusions

In this work, we not only conducted an HPLC analysis of the chemical composition of sweet basil but also demonstrated the enzymatic status and antioxidant status of both organic and conventional plants. Our findings showed that basil is a good source of bioactive compounds in our diet. Different cultivation methods can be useful as factors for increasing basil quality, such as polyphenol and carotenoid content. In the following three-year experiment, we showed the quantitative and qualitative status of organic and conventional basil, their reaction to environmental stresses, and the unique distribution of bioactive compounds in the leaves. Organic basil contained significantly more selected carotenoids, such as lutein (0.13 mg g^−^^1^ DW) and chlorophylls a and b (2.03 mg g^−^^1^ DW and 0.15 mg g^−^^1^ DW), than conventional basil (0.10 mg g^−^^1^ DW, 1.97 mg g^−^^1^ DW, and 0.10 mg g^−^^1^ DW), respectively. In this kind of aromatic plant, there was a tendency towards higher concentrations of polyphenols (5.24 mg g^−^^1^ DW) compared to conventional (4.16 mg g^−^^1^ DW) herbs. The obtained results presented in our experiment have high utility for basil producers. They can help create the environmental conditions that will result in the best quality of this popular aromatic plant. At the same time, the results obtained are very important for consumers, because they can help in choosing the right type of organic or conventional basil for daily use.

## Figures and Tables

**Figure 1 foods-13-00383-f001:**
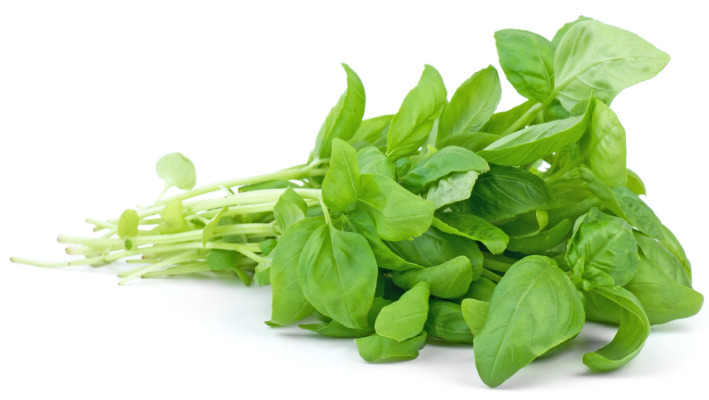
Plant of sweet basil used in the experiment for chemical composition evaluation.

**Figure 2 foods-13-00383-f002:**
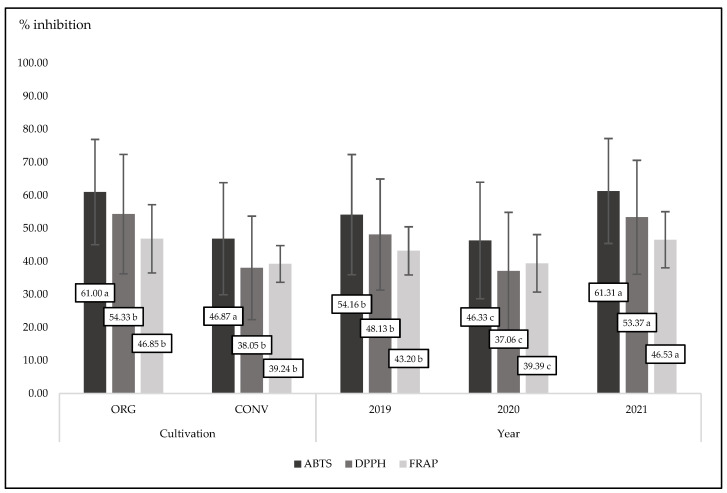
Mean values for antioxidant activity measured by three methods for organic and conventional basil cultivated in three experimental years; *p*-value (cultivation) < 0.0001; (year) < 0.0001 for three methods. Different letters are significantly different at the 5% level of probability.

**Figure 3 foods-13-00383-f003:**
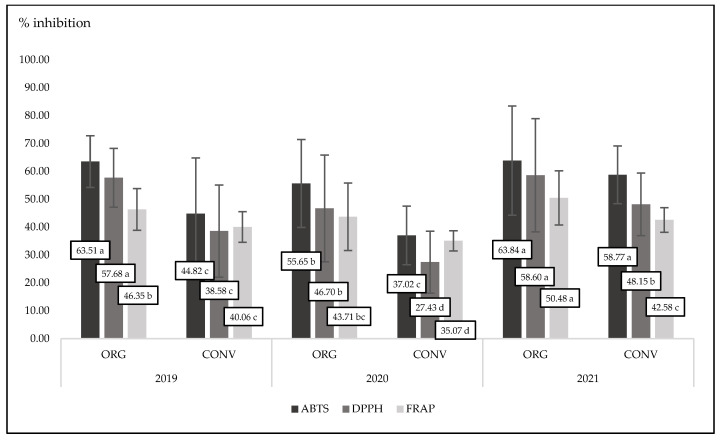
Interaction between experimental factors for antioxidant activity measured by three methods for organic and conventional basil cultivated in three experimental years; ABTS, *p* = 0.003; DPPH, *p* = n.s.; FRAP, *p* = n.s. Different letters are significantly different at the 5% level of probability.

**Figure 4 foods-13-00383-f004:**
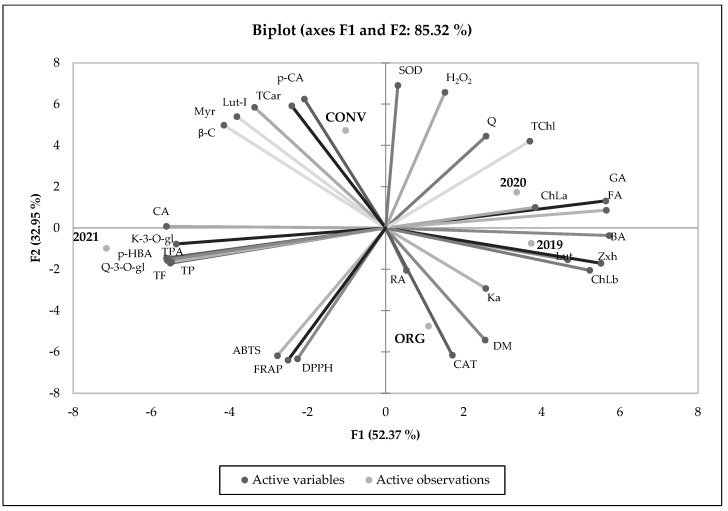
PCA analysis showing the relationship between the chemical composition and experimental years (2019, 2020, 2021) as well as the production system of sweet basil. DM, dry matter; GA, gallic acid; p-HBA, p-hydrobenzoic acid; CA, caffein acid; RA, rosmarinic acid; p-CA, p-coumaric acid; FA, ferulic acid; BA, benzoic acid; K-3-O-gl, kaempferol-3-O-glucoside; Myr, myrycetin; Lut-l, luteolin; Q, quercetin; Q-3-O-gl, quercetin-3-O-glucoside; Ka, keampferol; Lut, lutein; Zxh, zeaxanthin; β-C, beta-karotene; ChLa, chlorophyll a; ChLb, chlorophyll b; TP, total polyphenols; TPhA, total phenolics acids; TF, total flavonoids; TC, total carotenoids; TChL, total chlorophylls; H_2_O_2_, hydrogenperoxide; SOD, superoxide dismutase; CAT, catalase; ABTS, 2,2′-azino-bis(3-ethylbenzothiazoline-6-sulfonic acid); DPPH, 2,2-difenylo-1-pikrylohydrazyl; FRAP, ferric ion reducing antioxidant power.

**Figure 5 foods-13-00383-f005:**
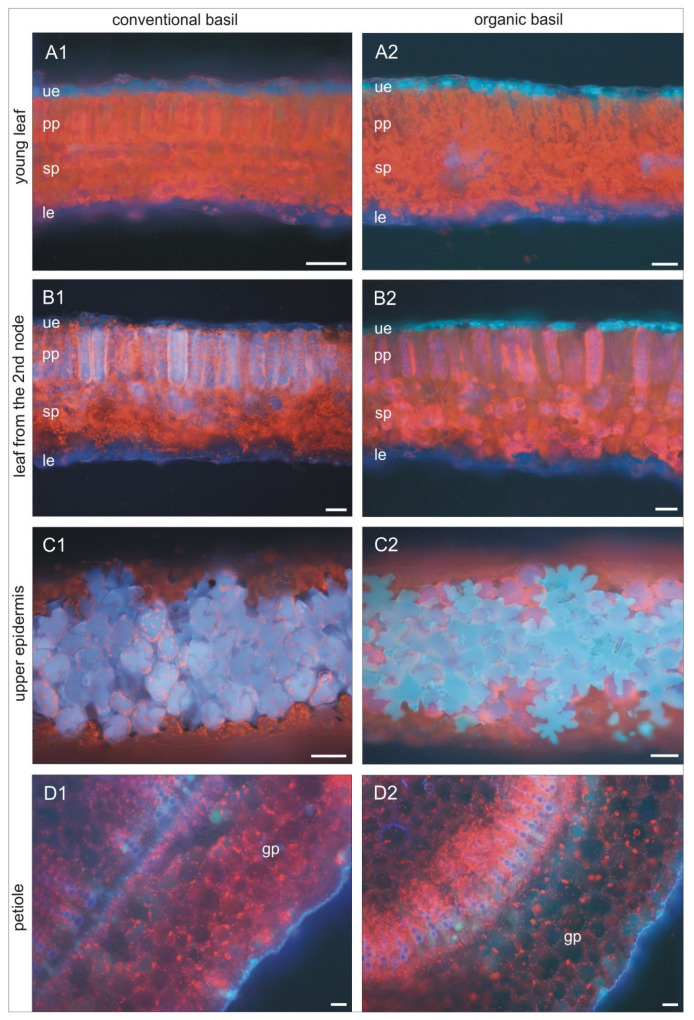
The microscopic picture of selected parts of basil leaves from organic and conventional production: (**A1**) conventional joung leaf, (**A2**) organic joung leaf, (**B1**) conventional leaf from 2nd node, (**B2**) organic leaf from 2nd node, (**C1**) conventional upper epidermis, (**C2**) organic upper epidermis, (**D1**) conventional petiole, and (**D2**) organic petiole, (pp) palisade parenchyma, (gp) ground parenchyma, (ue) upper epidermis, (le) lower epidermis, and (sp) spongy parenchyma; Scale bar = 5 micrometers.

**Table 1 foods-13-00383-t001:** The content of bioactive compounds in total (in mg g^−1^ DW) and dry matter (in g 100 g^−1^ FW) in basil from two production systems and in three experimental years.

Bioactive Compounds Groups/Experimental Combination	2019		2020		2021		*p*-Value
	ORG	CONV	ORG	CONV	ORG	CONV	C × Y
Polyphenols	4.34 ^1^ ± 0.93 ^a2^	4.16 ± 0.59 ^a^	4.71 ± 1.25 ^a^	4.33 ± 2.01 ^a^	6.66 ± 0.80 ^a^	6.57 ± 0.83 ^a^	N.S. ^3^
Phenolic acid	3.97 ± 0.56 ^a^	4.09 ± 0.58 ^a^	4.56 ± 1.11 ^a^	4.01 ± 2.24 ^a^	5.70 ± 0.90 ^a^	5.60 ± 0.80 ^a^	N.S.
Flavonoids	0.44 ± 0.25 ^a^	0.35 ± 0.06 ^a^	0.35 ± 0.13 ^a^	0.35 ± 0.07 ^a^	1.12 ± 0.10 ^a^	1.18 ± 0.13 ^a^	N.S.
Carotenoids	0.66 ± 0.16 ^b^	0.80 ± 0.07 ^ab^	0.75 ± 0.11 ^b^	0.91 ± 0.18 ^a^	0.45 ± 0.10 ^c^	1.19 ± 0.17 ^a^	<0.0001
Chlorophylls	2.25 ± 0.53 ^b^	2.64 ± 0.53 ^ab^	2.16 ± 0.31 ^b^	3.15 ± 0.45 ^a^	1.86 ± 0.73 ^c^	2.79 ± 1.11 ^a^	<0.0001
Dry matter	13.34 ± 3.59 ^a^	11.01 ± 2.32 ^b^	11.52 ± 2.95 ^b^	10.95 ± 4.59 ^c^	12.06 ± 5.15 ^a^	9.68 ± 4.91 ^c^	0.0045
Bioactive compounds groups/experimental combination	C (cultivation)		Y (year)			*p*-value	
	ORG	CONV	2019	2020	2021	C	Y
Polyphenols	5.24 ± 1.44 ^A^	4.99 ± 1.70 ^A^	4.25 ± 0.78 ^b^	4.52 ± 1.69 ^b^	6.62 ± 0.82 ^a^	N.S.	<0.0001
Phenolic acids	4.74 ± 1.14 ^A^	4.56 ± 1.59 ^A^	4.03 ± 0.57 ^b^	4.29 ± 1.79 ^b^	5.65 ± 0.85 ^a^	N.S.	0.0005
Flavonoids	0.64 ± 0.38 ^A^	0.61 ± 0.40 ^A^	0.40 ± 0.19 ^b^	0.35 ± 0.11 ^b^	1.15 ± 0.12 ^a^	N.S.	<0.0001
Carotenoids	0.72 ± 0.23 ^B^	0.85 ± 0.28 ^A^	0.73 ± 0.14 ^a^	0.83 ± 0.17 ^a^	0.82 ± 0.40 ^a^	0.002	N.S.
Chlorophylls	2.44 ± 0.76 ^A^	2.55 ± 0.81 ^A^	2.44 ± 0.57 ^a^	2.66 ± 0.63 ^a^	2.33 ± 1.05 ^a^	N.S.	N.S.
Dry matter	11.97 ± 4.17 ^A^	10.54 ± 4.15 ^B^	12.17 ± 3.24 ^a^	10.73 ± 3.86 ^b^	10.87 ± 5.17 ^b^	0.0001	<0.0001

^1^ Data are presented as the mean ± SD with ANOVA *p*-value; ^2^ Means in rows followed by the different letters are significantly different at the 5% level of probability (*p* < 0.05); ^3^ N.S., not statistically significant.

**Table 2 foods-13-00383-t002:** Enzymes activity, hydroxyperoxide concentration, and antioxidant activity in basil from two production systems and three experimental years.

Activity/Experimental Combination		H_2_O_2_	SOD	CAT
		µmol 100 mg^−1^	units mg^−1^	µmol H_2_O_2_ min^−1^ mg^−1^
2019	ORG	6.53 ± 0.74 ^b^	59.36 ± 6.00 ^a^	40.19 ± 3.47 ^a^
	CONV	8.88 ± 0.79 ^a^	79.83 ± 4.72 ^a^	20.38 ± 5.12 ^b^
2020	ORG	5.84 ± 0.83 ^bc^	58.21 ± 6.79 ^a^	40.10 ± 6.44 ^a^
	CONV	9.07 ± 0.16 ^a^	79.27 ± 3.80 ^a^	26.98 ± 3.26 ^b^
2021	ORG	5.76 ± 0.94 ^c^	55.26 ± 3.96 ^a^	37.53 ± 4.12 ^a^
	CONV	7.44 ± 0.46 a^b^	75.02 ± 3.07 ^a^	22.23 ± 6.86 ^b^
C (cultivation)	ORG	6.04 ± 0.91 ^B^	57.61 ± 5.69 ^B^	39.28 ± 5.00 ^A^
	CONV	8.47 ± 0.90 ^A^	78.04 ± 4.47 ^A^	23.19 ± 5.98 ^B^
Y (year)	2019	7.70 ± 1.40 ^a^	69.60 ± 11.57 ^a^	30.28 ± 10.83 ^a^
	2020	7.46 ± 1.72 ^a^	68.74 ± 11.88 ^a^	33.54 ± 8.31 ^a^
	2021	6.60 ± 1.12 ^b^	65.14 ± 10.50 ^b^	29.88 ± 9.52 ^b^
*p*-value	C	<0.0001	<0.0001	<0.0001
	Y	<0.0001	<0.0001	0.0004
	C × Y	<0.0001	N.S.	0.0035

Data are presented as the mean ± SD with ANOVA *p*-value; Means in columns followed by the different letters are significantly different at the level of probability (*p* < 0.05); N.S., not statistically significant.

**Table 3 foods-13-00383-t003:** The content of individual compounds (in mg g^−1^ DW) in basil from two production systems and in three experimental years.

Individual Compounds/Experimental Combination	2019		2020		2021		C (Cultivation)		Y (Year)			*p*-Value		
	ORG	CONV	ORG	CONV	ORG	CONV	ORG	CONV	2019	2020	2021	C	Y	C × Y
Gallic acid	0.35 ^1^ ± 0.06 ^a2^	0.33 ± 0.04 ^a^	0.34 ± 0.09 ^a^	0.35 ± 0.16 ^a^	0.11 ± 0.01 ^a^	0.12 ± 0.01 ^a^	0.27 ± 0.13 ^a^	0.27 ± 0.14 ^a^	0.34 ± 0.05 ^a^	0.34 ± 0.13 ^a^	0.11 ± 0.01 ^b^	N.S. ^3^	<0.0001	N.S.
p-Hydroxybenzoic	0.34 ± 0.07 ^b^	0.21 ± 0.03 ^c^	0.26 ± 0.11 ^c^	0.34 ± 0.19 ^b^	1.58 ± 0.13 ^a^	1.41 ± 0.13 ^a^	0.73 ± 0.62 ^a^	0.65 ± 0.55 ^a^	0.27 ± 0.08 ^b^	0.30 ± 0.16 ^b^	1.50 ± 0.15 ^a^	N.S.	<0.0001	0.018
Caffeic	0.14 ± 0.05 ^b^	0.13 ± 0.03 ^b^	0.19 ± 0.06 ^ab^	0.15 ± 0.05 ^b^	0.22 ± 0.08 ^ab^	0.33 ± 0.13 ^a^	0.18 ± 0.08 ^a^	0.20 ± 0.12 ^a^	0.13 ± 0.04 ^b^	0.17 ± 0.06 ^b^	0.28 ± 0.12 ^a^	N.S.	<0.0001	0.0141
Rosmarinic	2.87 ± 0.17 ^a^	1.88 ± 0.22 ^a^	2.38 ± 0.32 ^a^	2.52 ± 0.16 ^a^	2.28 ± 0.19 ^a^	2.01 ± 0.14 ^a^	2.18 ± 0.31 ^a^	2.12 ± 0.41 ^a^	2.12 ± 0.23 ^a^	2.27 ± 0.41 ^a^	2.21 ± 0.24 ^a^	N.S.	N.S.	N.S.
p-Coumaric	0.06 ± 0.04 ^c^	0.23 ± 0.04 ^a^	0.10 ± 0.07 ^b^	0.15 ± 0.05 ^b^	0.04 ± 0.00 ^c^	0.29 ± 0.02 ^a^	0.07 ± 0.05 ^b^	0.22 ± 0.08 ^a^	0.14 ± 0.09 ^a^	0.13 ± 0.09 ^a^	0.16 ± 0.12 ^a^	<0.0001	N.S.	<0.0001
Ferulic	0.054 ± 0.002 ^a^	0.063 ± 0.001 ^a^	0.054 ± 0.002 ^a^	0.053 ± 0.002 ^a^	0.025 ± 0.001 ^a^	0.014 ± 0.001 ^a^	0.044 ± 0.002 ^a^	0.043 ± 0.003 ^a^	0.058 ± 0.001 ^a^	0.053 ± 0.002 ^a^	0.020 ± 0.001 ^b^	N.S.	<0.0001	N.S.
Benzoic	0.14 ± 0.11 ^a^	0.12 ± 0.03 ^a^	0.11 ± 0.06 ^a^	0.13 ± 0.08 ^a^	0.09 ± 0.01 ^a^	0.04 ± 0.01 ^a^	0.11 ± 0.08 ^a^	0.10 ± 0.06 ^a^	0.13 ± 0.08 ^a^	0.12 ± 0.07 ^a^	0.07 ± 0.03 ^b^	N.S.	0.025	N.S.
Kaempferol-3-O-glucoside	0.028 ± 0.001 ^a^	0.037 ± 0.001 ^a^	0.027 ± 0.001 ^a^	0.025 ± 0.001 ^a^	0.043 ± 0.001 ^a^	0.044 ± 0.001 ^a^	0.033 ± 0.001 ^a^	0.035 ± 0.001 ^a^	0.032 ± 0.001 ^b^	0.026 ± 0.001 ^c^	0.044 ± 0.001 ^a^	N.S.	<0.0001	N.S.
Myricetin	0.038 ± 0.010 ^a^	0.058 ± 0.010 ^a^	0.054 ± 0.030 ^a^	0.059 ± 0.030 ^a^	0.034 ± 0.001 ^a^	0.106 ± 0.050 ^a^	0.042 ± 0.020 ^a^	0.075 ± 0.040 ^a^	0.048 ± 0.010 ^c^	0.056 ± 0.030 ^b^	0.070 ± 0.050 ^a^	N.S.	0.0007	N.S.
Luteolin	0.017 ± 0.001 ^b^	0.024 ± 0.001 ^a^	0.021 ± 0.001 ^a^	0.024 ± 0.010 ^a^	0.019 ± 0.001 ^ab^	0.030 ± 0.001 ^a^	0.019 ± 0.001 ^b^	0.026 ± 0.010 ^a^	0.020 ± 0.010 ^a^	0.022 ± 0.010 ^a^	0.024 ± 0.010 ^a^	<0.0001	N.S.	0.046
Quercetin	0.039 ± 0.020 ^a^	0.042 ± 0.020 ^a^	0.037 ± 0.020 ^a^	0.032 ± 0.010 ^ab^	0.019 ± 0.001 ^b^	0.045 ± 0.001 ^a^	0.032 ± 0.020 ^a^	0.040 ± 0.010 ^a^	0.041 ± 0.020 ^a^	0.035 ± 0.020 ^a^	0.032 ± 0.010 ^a^	0.036	N.S.	0.005
Quercetin-3-O-glucoside	0.119 ± 0.120 ^b^	0.118 ± 0.080 ^b^	0.070 ± 0.060 ^c^	0.073 ± 0.060 ^c^	0.781 ± 0.050 ^a^	0.746 ± 0.080 ^a^	0.324 ± 0.034 ^a^	0.312 ± 0.032 ^a^	0.119 ± 0.010 ^b^	0.071 ± 0.060 ^c^	0.764 ± 0.070 ^a^	N.S.	<0.0001	<0.0001
Kaempferol	0.029 ± 0.001 ^a^	0.032 ± 0.010 ^a^	0.021 ± 0.001 ^b^	0.022 ± 0.001 ^b^	0.028 ± 0.001 ^a^	0.017 ± 0.001 ^c^	0.026 ± 0.010 ^a^	0.023 ± 0.010 ^a^	0.030 ± 0.010 ^a^	0.021 ± 0.001 ^b^	0.022 ± 0.010 ^b^	N.S.	0.0004	0.007
Lutein	0.13 ± 0.02 ^a^	0.11 ± 0.06 ^a^	0.17 ± 0.05 ^a^	0.12 ± 0.07 ^a^	0.10 ± 0.01 ^a^	0.07 ± 0.02 ^a^	0.13 ± 0.04 ^a^	0.10 ± 0.05 ^B^	0.12 ± 0.04 ^b^	0.15 ± 0.06 ^a^	0.09 ± 0.02 ^b^	0.022	0.0032	N.S.
Zeaxanthin	0.027 ± 0.001 ^a^	0.027 ± 0.010 ^a^	0.027 ± 0.001 ^a^	0.024 ± 0.010 ^a^	0.023 ± 0.001 ^a^	0.020 ± 0.001 ^a^	0.026 ± 0.001 ^a^	0.023 ± 0.010 ^a^	0.027 ± 0.010 ^a^	0.026 ± 0.001 ^a^	0.021 ± 0.001 ^a^	N.S.	N.S.	N.S.
Beta-carotene	0.52 ± 0.15 ^b^	0.40 ± 0.03 ^c^	0.57 ± 0.16 ^b^	0.40 ± 0.07 ^c^	0.20 ± 0.02 ^d^	0.94 ± 0.02 ^a^	0.43 ± 0.21 ^b^	0.58 ± 0.28 ^a^	0.46 ± 0.12 ^a^	0.49 ± 0.15 ^a^	0.57 ± 0.39 ^a^	0.0004	N.S.	<0.0001
Chlorophyll a	2.05 ± 0.46 ^ab^	1.82 ± 0.47 ^b^	2.63 ± 0.28 ^a^	1.78 ± 0.35 ^b^	1.41 ± 0.46 ^c^	2.30 ± 0.82 ^a^	2.03 ± 0.64 ^a^	1.97 ± 0.63 ^a^	1.93 ± 0.48 ^a^	2.21 ± 0.53 ^a^	1.85 ± 0.80 ^a^	N.S.	N.S.	<0.0001
Chlorophyll b	0.24 ± 0.07 ^a^	0.15 ± 0.05 ^ab^	0.16 ± 0.05 ^ab^	0.11 ± 0.02 ^b^	0.06 ± 0.00 ^c^	0.05 ± 0.01 ^c^	0.15 ± 0.09 ^a^	0.10 ± 0.05 ^b^	0.19 ± 0.08 ^a^	0.13 ± 0.04 ^b^	0.06 ± 0.01 ^c^	0.0002	<0.0001	0.044

^1^ Data are presented as the mean ± SD with ANOVA *p*-value; ^2^ Means in rows followed by the different letters are significantly different at the 5% level of probability (*p* < 0.05); ^3^ N.S., not statistically significant.

## Data Availability

Data is contained within the article or Appendix A.

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
