# Peer review of "A Long-Term Study on Chemical Compounds and Their Location in Sweet Basil Leaves from Organic and Conventional Producers"

_foods, 2024, doi:10.3390/foods13030383_

Round 1
Reviewer 1 Report
Comments and Suggestions for Authors
Hallmann et al. have aimed to make a long-term study on chemical compounds and their location in sweet basil leaves from organic and conventional producers. This study contains some informative data and should be of interest in this field. However, the following points need to be addressed:
1. The language needs a moderate revision throughout the manuscript.
2. The title should be modified as “A long-term study on chemical compounds and their location in sweet basil leaves from organic and conventional producers”.
3. The keyword ‘chlorophylls’ should be added under the keywords section.
4. The objectives in the last line of the ‘introduction’ should be expanded to clearly state the complete objectives with more details.
5. All the purchase details of chemicals/reagents and instruments/equipment/software/kits should be provided as state, city, and country in the case of USA as well as city and country in the case of other countries. Also, the authors can just mention the company name for the second instance.
6. Doublecheck all the equations/formula under the section ‘materials & methods’ for correctness.
7. The reference citations in text are mixed up in style. Please check lines 259, 267, 289, 300, 310, etc.
8. The data obtained in this study should be subjected to principal component analysis for getting useful information on correlations between different parameters studied.
9. Table 1 caption – compounds’ should be corrected as compounds.
10. Line 345, 369 – ‘in plants’ should be corrected as ‘in basil’
11. In the table footnotes and figure captions, the explanation for NS should be ‘not statistically significant’.
12. Tables 1, 2 and 3 & Figure 2 – Why there are mix-up of small and capital letters for denoting statistically significant letters? If it is meaningful, the authors should explain what treatments the small letters compare and the capital letters compare.
13. Figure 4 – these letters/abbreviations used in microscopic pictures should be explained in the figure caption. That is A1, A2, B1, B2, C1, C2, D1, D2, ue, pp, sp, le, gp.
14. The peak 3 in Figure S1 A and B doesn’t seem to be resolved properly, as it looks like a combination of two peaks.
15. Why the peak numbers are not given in order in the chromatograms A and B in Figure S2.
16. Is it zeaxanthin or zeaxanthinid?
Comments on the Quality of English LanguageModerate editing of English language required
Author Response
Replies to Reviewer no. 1
Thank you very much for the review and for your positive recommendation to publish our manuscript in the Foods journal. Below you can find answers for your comments and suggestions:
Comment 1: “ The language needs a moderate revision throughout the manuscript. ”
Authors’ response: According to Reviewer suggestion manuscript was checked by professional language translation company.
Comment 2: “The title should be modified as “A long-term study on chemical compounds and their location in sweet basil leaves from organic and conventional producers ”
Authors’ response: According to Reviewer suggestion the title was modified and changed.
Comment 3: “The keyword ‘chlorophylls’ should be added under the keywords section ”
Authors’ response: According to Reviewer suggestion word “chlorophyll” was added into keywords
Comment 4: “ The objectives in the last line of the ‘introduction’ should be expanded to clearly state the complete objectives with more details”
Authors’ response: According to Reviewer suggestion correction of last line Introduction section was done (lines 110-112)
Comment “All the purchase details of chemicals/reagents and instruments/equipment/software/kits should be provided as state, city, and country in the case of USA as well as city and country in the case of other countries. Also, the authors can just mention the company name for the second instance.”
Authors’ response: According to Reviewer suggestion properly correction to both sub-section “Chemicals and reagents” and “Equipment” were done
Comment 6: “Doublecheck all the equations/formula under the section ‘materials & methods’ for correctness ”
Authors’ response: According to Reviewer suggestion all equations and mathematical formulas were checked carefully. All mistakes are corrected.
Comment 7: “The reference citations in text are mixed up in style. Please check lines 259, 267, 289, 300, 310, etc.”
Authors’ response: According to Reviewer suggestion all references in text were checked and unified into one style accepted by Foods journal template.
Comment 8: “The data obtained in this study should be subjected to principal component analysis for getting useful information on correlations between different parameters studied ”
Authors’ response: According to Reviewer suggestion PCA analysis of obtained data was added to manuscript text.
Comment 9: “Table 1 caption – compounds’ should be corrected as compounds ”
Authors’ response: According to Reviewer suggestion correction was done.
Comment 10: “Line 345, 369 – ‘in plants’ should be corrected as ‘in basil’”
Authors’ response: According to Reviewer suggestion correction was done.
Comment 11: “In the table footnotes and figure captions, the explanation for NS should be ‘not statistically significant’ ”
Authors’ response: According to Reviewer suggestion correction was done.
Comment 12: “Tables 1, 2 and 3 & Figure 2 – Why there are mix-up of small and capital letters for denoting statistically significant letters? If it is meaningful, the authors should explain what treatments the small letters compare and the capital letters compare. ”
Authors’ response: According to Reviewer suggestion all letters for homogenous groups were unified into one style.
Comment 13: “Figure 4 – these letters/abbreviations used in microscopic pictures should be explained in the figure caption. That is A1, A2, B1, B2, C1, C2, D1, D2, ue, pp, sp, le, gp.”
Authors’ response: According to Reviewer suggestion missing letters/abbreviation for Figure 5 were added into figure caption.
Comment 14: “The peak 3 in Figure S1 A and B doesn’t seem to be resolved properly, as it looks like a combination of two peaks.”
Authors’ response: According to Reviewer suggestion detailed revision of peaks identification were done. Thanks due to, wrong identification of caffeic acid and rosmarinic acid were detected. As Reviewer points one (the highest pick) was really combination of two peaks caffeic and rosmarinic acid. Because in 2022, when chemical analysis took a place, our team not have rosmarinic acid standards. Moreover we wrong identified those two peaks. After Reviewer suggestion we purchased chemical standard and again identified (after slightly method modification) caffeic and rosmarinic acid.
Comment 15: “Why the peak numbers are not given in order in the chromatograms A and B in Figure S2.”
Authors’ response: According to Reviewer suggestion subsequent peaks appearing in the chromatogram were numbered with consecutive numbers.
Comment 16: “Is it zeaxanthin or zeaxanthinid? ”
Authors’ response: According to Reviewer suggestion wrong word “zeaxanthinid” was corrected into “zeaxanthin”
Reviewer 2 Report
Comments and Suggestions for Authors
This study describes the effect of conventional and organic farming on the phytochemicals composition and distribution, antioxidant potential and antioxidant enzyme status in Basil leaves. The topic of this study is quite interesting, however, the presented issues seem more suitable for the journal covering the scope of agricultural or plant science (for example Agriculture or Plants). Despite the study being conducted on culinary herb, the potential nutritional aspects should be highlighted for the “Foods” journal. The abstract reflects the content of the work but it needs some minor corrections. In general, the introduction provides a basic background of the study. The aim of the study was included in the introduction. The experiments are in general properly designed and described in sufficient detail (necessary references were added) to reproduce analyses. A wide range of performed phytochemical analyses is laudable, especially in terms of the pro-health potential of basil leaves. Nevertheless, some analyses like hydrogen peroxide level determination, antioxidant enzyme activity and microscopic analyses, despite they are showing changes in plant metabolism are valuable from plant physiology and seem to have limited importance for food science. In addition, some methodological aspects should be clarified (details described below). The modes of results presentation are proper. The discussion is supported by the results, however, in the discussion section authors should more extensively refer to nutritional/nutraceutical aspects of the study to emphasise the significance of this study for food science. The conclusion summarizes the findings and highlights the utilitary application for basil producers; however, it will be valuable to emphasize the significance of obtained results for potential consumers. Authors should let check the manuscript by a native speaker to exclude some grammatical and syntax errors. Due to some shortcomings influencing the quality of the manuscript I recommend major revision. Authors should thoroughly revise the manuscript and provide necessary clarifications.
Detailed suggestions:
Abstract
The abstract needs revision and some corrections to improve its clarity:
Line 21: „Unique chemical were also estimated
by fluorescence microscopy.” Avoid duplication of „also” (previous phrase). The term ”unique” seems speculative. Consider: Fluorescence microscopy was used for the determination of compounds locations in the basil leaves.
Line 23: remove “largest” (not significant and speculative information)
Line 23-24: “The results showed that the chemical structures of organic and conventional basil leaves are different.” Replace „structures” with „profiles”. The chemical structures of compounds are the same.
Line 26-28: “Organic basil contained significantly more dry matter (11.97 g 100 g-1 FW) compare to conventional one (10.54 g 100g-1 FW) and showed a higher tendency for total phenolic (5.24 mg g -1 DW) concentrations than conventional basil (4.99 mg g-1 DW).” This phrase is unclear, especially „a higher tendency for total phenolic (5.24 mg g -1 DW) concentrations” Maybe, „a higher tendency for total phenolic compounds ( 5.24 mg g -1 DW) accumulation.” or similarly?
Line 28-32: “The higher bioactive compound…” The second part of this phrase „ as well catalase activity (39 μ mol H2O2 min-1 mg-1) in organic compare to conventional (23.19 μ mol H2O2 min-1 mg-1) of examined basil” is unclear. Divide it into 2 sentences or rephrase this part.
Line 32-33: In the conclusion sentence refer also to nutritional or nutraceutical aspects of the study.
Introduction
Line 45: “gallic, caffeic, chlorogenic, hydroxybenzoic, vanillic, ferulic, and trans-cinaminic [3],[4].” Wy rosmarinic acid was not mentioned? In many studies, it is considered one of the main phenolic secondary metabolites of basil. It was also mentioned in the work [3].
Line 46, 74 etc. : replace “[3],[4]” by [3,4]; „[19]-[22]” by [19-22]. Check it through the manuscript.
Line 67-68: Remove „Basil is a herb, which contains a lot of bioactive compounds from phenolic and carotenoids groups.” The same information was provided in line 43 „Basil is plant-rich in different bioactive compounds..” Consider rephrasing the second sentence: „The quality and quantity profile of bioactive compounds is depended from plant genetic, but from farm management as well.”
Line 111: If possible, add information about the variety of basil. Was the same variety used in conventional and organic farming?
Line 119: „Seeds were watered by water” remove „by water”. „Watered” indicates that water was used.
Materials and Methods
Line 87: Add „reagents” after the „following”
Line 151, 156, 170 etc. Add subsections i.e. 2.3.1, 2.3.2 ….
Line 158: „..and 50 mL of deionized water was added.” Why authors decide to use water for polyphenol extraction? In this case, only the most polar compounds ​were privileged for extraction. Methanol, ethanol or its water solutions seem more adequate for the isolation of less polar phenolic compounds like flavonoids.
Line 232, 233: replace “oC” with „°C”
Line 236: “-1” should be in superscript
Line 278: „(%) = [(A0 − A1)/A0] × 100” Was the same equation used for ABTS, DPPH and FRAP assays? What was A0 for FRAP determination?
Results and discussion
Authors should in discussion section more eextensively reffer to nutritional/nutracetical aspects of the study to emphasise the significance of this study for food science e.g Why the determination of the contents of phytochemicals and antioxidant activity is important from a nutritional/nutraceutical point of view? Why farming method is important from a nutritional/nutraceutical point of view? Authors in line 392 stated that “Basil is a good source of lutein” describe what is pro-helath potential of lutein… etc.
Line 333: double ]]
Line 361, Table 2: The main drawback of polyphenols profiling by HPLC is lack of the rosmarinic acid determination.
Line 400: “Enzymatic status of plants” Authors should consider moving this paragraph (also in materials and methods) before results for phytochemical analysis to make more systematical order as the contents of phytochemicals are the result of metabolism changes influenced by environmental factors including cultivation method (reflected by H2O2 level and antioxidant enzymes activity)
Table 2: remove the empty row.
Table 3: Correct the table placement. The text covers with line numbering.
Line 468 – Figure 2: „ mrthods” Description of Y axis is „% activity”, but in lines 265 and 270 authors stated that percentage inhibition was calculated. Clarify or unify it.
Conclusions:
586-587: „Obtained results presented in our experiment have a high utility application for basil producers. They can help them create the environmental conditions that will result in the best quality of this popular aromatic plant.” Speculate also the significance of obtained results for potential consumers.
Figure S1: Chromatograms show that some peaks (in particular peaks 6,7,10,12) look rather poorly resolved, which could influence the peaks integration and quantification of corresponding compounds. Authors should more precisely choose separation conditions. Chromatogram acquisition wavelength should be added.
Author Response
Replies to Reviewer no. 2
Thank you very much for the review and for your positive recommendation to publish our manuscript in the Foods journal. Below you can find answers for your comments and suggestions:
Comment 1: “Line 21: „Unique chemical were also estimated
by fluorescence microscopy.” Avoid duplication of „also” (previous phrase). The term ”unique” seems speculative. Consider: Fluorescence microscopy was used for the determination of compounds locations in the basil leaves. “
Authors’ response: According to Reviewer suggestion properly correction were done.
Comment 2: “Line 23: remove “largest” (not significant and speculative information) “
Authors’ response: According to Reviewer suggestion properly correction was done
Comment 3: “Line 23-24: “The results showed that the chemical structures of organic and conventional basil leaves are different.” Replace „structures” with „profiles”. The chemical structures of compounds are the same“
Authors’ response: According to Reviewer suggestion properly correction was done
Comment 4: “Line 26-28: “Organic basil contained significantly more dry matter (11.97 g 100 g-1 FW) compare to conventional one (10.54 g 100g-1 FW) and showed a higher tendency for total phenolic (5.24 mg g -1 DW) concentrations than conventional basil (4.99 mg g-1 DW).” This phrase is unclear, especially „a higher tendency for total phenolic (5.24 mg g -1 DW) concentrations” Maybe, „a higher tendency for total phenolic compounds ( 5.24 mg g -1 DW) accumulation.” or similarly? “
Authors’ response: According to Reviewer suggestion properly correction were done
Comment 5: “Line 28-32: “The higher bioactive compound…” The second part of this phrase „ as well catalase activity (39 μ mol H2O2 min-1 mg-1) in organic compare to conventional (23.19 μ mol H2O2 min-1 mg-1) of examined basil” is unclear. Divide it into 2 sentences or rephrase this part.“
Authors’ response: According to Reviewer suggestion properly correction was done
Comment 6: “Line 32-33: In the conclusion sentence refer also to nutritional or nutraceutical aspects of the study. “
Authors’ response: According to Reviewer suggestion properly correction were done
Comment 7: “Line 45: “gallic, caffeic, chlorogenic, hydroxybenzoic, vanillic, ferulic, and trans-cinaminic [3],[4].” Wy rosmarinic acid was not mentioned? In many studies, it is considered one of the main phenolic secondary metabolites of basil. It was also mentioned in the work [3].“
Authors’ response: According to Reviewer suggestion detailed revision of picks identification were done. Thanks due to, wrong identification of caffeic acid and rosmarinic acid were detected. As Reviewer points one (the highest pick) was really combination of two picks caffeic and rosmarinic acid. Because in 2022, when chemical analysis took a place, our team not have rosmarinic acid standards. Moreover we wrong identified those two picks. After Reviewer suggestion we purchased chemical standard and again identified (after slightly method modification) caffeic and rosmarinic acid.
Comment 8: “Line 46, 74 etc. : replace “[3],[4]” by [3,4]; „[19]-[22]” by [19-22]. Check it through the manuscript.“
Authors’ response: According to Reviewer suggestion properly correction were done
Comment 9: “Line 67-68: Remove „Basil is a herb, which contains a lot of bioactive compounds from phenolic and carotenoids groups.” The same information was provided in line 43 „Basil is plant-rich in different bioactive compounds..” Consider rephrasing the second sentence: „The quality and quantity profile of bioactive compounds is depended from plant genetic, but from farm management as well.”
Authors’ response: According to Reviewer suggestion properly correction were done
Comment 10: “Line 111: If possible, add information about the variety of basil. Was the same variety used in conventional and organic farming “
Authors’ response: According to Reviewer suggestion properly correction was done. Authors confirm, that for all experiment combinations Genovense cv. was used.
Comment 11: “ Line 119: „Seeds were watered by water” remove „by water”. „Watered” indicates that water was used. “
Authors’ response: According to Reviewer suggestion properly correction was done.
Comment 12: “Line 87: Add „reagents” after the „following “
Authors’ response: According to Reviewer suggestion properly correction was done.
Comment 13: “Line 151, 156, 170 etc. Add subsections i.e. 2.3.1, 2.3.2 ….“
Authors’ response: According to Reviewer suggestion properly correction was done.
Comment 14: “ Line 158: „..and 50 mL of deionized water was added.” Why authors decide to use water for polyphenol extraction? In this case, only the most polar compounds ​were privileged for extraction. Methanol, ethanol or its water solutions seem more adequate for the isolation of less polar phenolic compounds like flavonoids.“
Authors’ response: The Authors want to apologize. It was a mistake. A mixture of water and methanol was used to extract polyphenolic compounds, of course. Properly correction was done in manuscript text.
Comment 15: “ Line 232, 233: replace “oC” with „°C”“
Authors’ response: According to Reviewer suggestion properly correction was done
Comment 16: “ Line 278: „(%) = [(A0 − A1)/A0] × 100” Was the same equation used for ABTS, DPPH and FRAP assays? What was A0 for FRAP determination? “
Authors’ response: All equations and mathematical formulas were carefully checked and corrected in manuscript text.
Comment 17: “ Authors should in discussion section more eextensively reffer to nutritional/nutracetical aspects of the study to emphasise the significance of this study for food science e.g Why the determination of the contents of phytochemicals and antioxidant activity is important from a nutritional/nutraceutical point of view? Why farming method is important from a nutritional/nutraceutical point of view? Authors in line 392 stated that “Basil is a good source of lutein” describe what is pro-helath potential of lutein… etc.“
Authors’ response: According to Reviewer suggestion properly correction were done
Comment 18: “ Line 333: double ]]“
Authors’ response: According to Reviewer suggestion properly correction was done
Comment 19: “ Line 361, Table 2: The main drawback of polyphenols profiling by HPLC is lack of the rosmarinic acid determination “
Authors’ response: Authors explain this mistake in response to comment no. 7. Rosmarinic acid and caffeic acid were wrong identified. Now this mistake was corrected.
Comment 20: “ Line 400: “Enzymatic status of plants” Authors should consider moving this paragraph (also in materials and methods) before results for phytochemical analysis to make more systematical order as the contents of phytochemicals are the result of metabolism changes influenced by environmental factors including cultivation method (reflected by H2O2 level and antioxidant enzymes activity) “
Authors’ response: According to Reviewer suggestion properly correction was done
Comment 21: “Table 2: remove the empty row.“
Authors’ response: According to Reviewer suggestion properly correction was done
Comment 22: “ Table 3: Correct the table placement. The text covers with line numbering. “
Authors’ response: According to Reviewer suggestion properly correction was done
Comment 23: “ Line 468 – Figure 2: „ mrthods” Description of Y axis is „% activity”, but in lines 265 and 270 authors stated that percentage inhibition was calculated. Clarify or unify it.“
Authors’ response: Authors want to apologise for mistake. Properly correction “% activity” into “% of inhibition” was done.
Comment 24: “ 586-587: „Obtained results presented in our experiment have a high utility application for basil producers. They can help them create the environmental conditions that will result in the best quality of this popular aromatic plant.” Speculate also the significance of obtained results for potential consumers “
Authors’ response: According to Reviewer suggestion properly correction was done
Comment 25: “Figure S1: Chromatograms show that some peaks (in particular peaks 6,7,10,12) look rather poorly resolved, which could influence the peaks integration and quantification of corresponding compounds. Authors should more precisely choose separation conditions. Chromatogram acquisition wavelength should be added.“
Authors’ response: The presented chromatogram is a illustrative figure. When identifying peaks, the Authors perform manual peak integration. Unfortunately, it is not possible to show the line cutting off the peaks in the chromatogram. However, based on the cut-off line, the peaks are separated and calculated.
Reviewer 3 Report
Comments and Suggestions for Authors
Dear authors
I have read very carefully the article presented by you and in the following I will formulate a series of observations that will justify the choice of my answer.
Favorable arguments
The research is correctly conducted, in accordance with the working hypothesis that aims to evaluate this plant in 3 consecutive years, in organic conditions and in conventional conditions, from the point of view of the phytochemical composition and from the point of view of the antioxidant action.
Weaknesses
A simple search for the word basil in scientific databases yields many study results. I wonder what is the real novelty of this study? The authors conclude that, except for the superior quality of the organic product (known fact), the year influences the determined parameters. In my opinion, the interpretation would be scientifically more rigorous if the authors linked this to the climatic conditions of the respective years, in the conditions in which the same cultivation method is used every year.
My recommendation is to publish with major revisions. In my opinion, the discussion of the results, which should be discussed on -long term study- according to the title, does not make substantial contributions to -create the environmental conditions that will result in the best quality of this popular aromatic plant- as the authors stipulate in the conclusions
Comments on the Quality of English LanguageThe article is easy to read from the point of view of the English language. However, I noticed a series of language errors (phrases without subject and predicate), which is why I suggest to the authors a careful check of the English language used
Author Response
Replies to Reviewer no. 3
Thank you very much for the review and for your positive recommendation to publish our manuscript in the Foods journal. Below you can find answers for your comments and suggestions:
Comment 1: A simple search for the word basil in scientific databases yields many study results. I wonder what is the real novelty of this study? The authors conclude that, except for the superior quality of the organic product (known fact), the year influences the determined parameters. In my opinion, the interpretation would be scientifically more rigorous if the authors linked this to the climatic conditions of the respective years, in the conditions in which the same cultivation method is used every year.
Authors’ response: According to Reviewer suggestion more elements were added for discussion section to show the effect of climatic conditions (temperature and light) on the examined basil parameters.
Comment 2: My recommendation is to publish with major revisions. In my opinion, the discussion of the results, which should be discussed on -long term study- according to the title, does not make substantial contributions to -create the environmental conditions that will result in the best quality of this popular aromatic plant- as the authors stipulate in the conclusions
Authors’ response: According to Reviewer suggestion and recommendations more elements of long-term study were added to discussion section. As the Authors pointed in scientific data base is only one two-years experiment with basil. Therefore other fruits and vegetables were used for comparison of obtained results.
Comment 3: The article is easy to read from the point of view of the English language. However, I noticed a series of language errors (phrases without subject and predicate), which is why I suggest to the authors a careful check of the English language used.
Authors’ response: According to Reviewer suggestion manuscript text was checked by a professional language translator company.
Reviewer 4 Report
Comments and Suggestions for Authors
A good article with a lot of experimental data that has been statistically correctly interpreted. However, some minor improvements would be desirable:
The introduction should be rewritten as there is some repeating information that could be part of a single paragraph,
- one about Culinar herb line 38 and basil is a herb line 67 which would include all the information about basil as herb
- one about its anticancer activity line 54, 60, 62, 64 and 66
lines 106-107: usually each reagent is told where it comes from or at least is listed by supplier and not as written in the text.
Perhaps an equipment sub-chapter would be useful: spectrometer, HLPC and other equipments
will be listed here, to avoid repetition in text.
lines 122-123. Sub-chapter 2.2.: A brief description of Humvit bio, Humiplant, Algaplant, Humvit bio universal and Ziołovit universal, Agrolinija-S, Basofoliar 2.0 should be made so that the reader knows what it contains.
line 157: Uniform method name Folin-Ciocâlteu method nad line 162 Folin-Ciocalteu reagent
lines 199-200: Please specify clearly the composition of mobile phase A and mobile phase B
line 205: Please be more specific about the external standards
lines 235-239: the stationary phase must be added to the HPLC method for chlorophylls and carotenoids, togheter with information about external standards
line 331-332 or 432-433: in text, sometimes the comparison with other plants that are not part of the experiment seems odd. Perhaps a comparison first with literature data on basil and then with other plants would be more useful.
Comments on the Quality of English Language
Perhaps the authors could bring more clarity of expression.
Author Response
Replies to Reviewer no. 4
Thank you very much for the review and for your positive recommendation to publish our manuscript in the Foods journal. Below you can find answers for your comments and suggestions:
Comment 1: one about Culinar herb line 38 and basil is a herb line 67 which would include all the information about basil as herb
Authors’ response: According to Reviewer suggestion repeating information about culinary properties of basil was removing from the manuscript text.
Comment 2: one about its anticancer activity line 54, 60, 62, 64 and 66
Authors’ response: According to Reviewer suggestion repeating information about anticancer activity of basil was removing from the manuscript text.
Comment 3: lines 106-107: usually each reagent is told where it comes from or at least is listed by supplier and not as written in the text.
Authors’ response: According to Reviewer suggestion sub-section “Chemical and reagents” was corrected. Now each listed reagents used in experiment has name of supplier, city and country of production).
Comment 4: Perhaps an equipment sub-chapter would be useful: spectrometer, HLPC and other equipments will be listed here, to avoid repetition in text.
Authors’ response: According to Reviewer suggestion sub-section “Equipment” was created.
Comment 5: lines 122-123. Sub-chapter 2.2.: A brief description of Humvit bio, Humiplant, Algaplant, Humvit bio universal and Ziołovit universal, Agrolinija-S, Basofoliar 2.0 should be made so that the reader knows what it contains.
Authors’ response: According to Reviewer suggestion detail characterization of composition fertilizers used in time of basil cultivation were added in Table S1 (Supplementary material)
Comment 6: line 157: Uniform method name Folin-Ciocâlteu method nad line 162 Folin-Ciocalteu reagent
Authors’ response: According to Reviewer suggestion method name Folin-Ciocâlteu was unified in whole manuscript text.
Comment 7: lines 199-200: Please specify clearly the composition of mobile phase A and mobile phase B
Authors’ response: According to Reviewer suggestion all missing details about mobile phase A and B was added to manuscript text.
Comment 8: line 205: Please be more specific about the external standards
Authors’ response: According to Reviewer suggestion all external standards were listed in manuscript text. All names of suppliers, country and city of origin were given into sub-section “Chemical and reagents”
Comment 9: lines 235-239: the stationary phase must be added to the HPLC method for chlorophylls and carotenoids, togheter with information about external standards
Authors’ response: According to Reviewer suggestion stationary phase (column parameters) and all external standards were listed in manuscript text. All names of suppliers, country and city of origin were given into sub-section “Chemical and reagents”.
Comment 10: line 331-332 or 432-433: in text, sometimes the comparison with other plants that are not part of the experiment seems odd. Perhaps a comparison first with literature data on basil and then with other plants would be more useful.
Authors’ response: According to Reviewer suggestion in pointed manuscript sections more comparison to other experiment with basil as well as other fruits (sour cherry and apricots) were added into manuscript text.
Round 2
Reviewer 1 Report
Comments and Suggestions for Authors
The authors have satisfactorily addressed all the comments raised by reviewers and substantially improved the overall quality of the article. Therefore, I recommend accepting this article for publication in 'Foods' journal.
Comments on the Quality of English LanguageMinor editing of English language required
Reviewer 2 Report
Comments and Suggestions for Authors
The manuscript was corrected accordingly and sufficient explanations were provided. In my opinion, the current version of the manuscript meets the basic journal requirements.